# MULTI-AGENT CROSS-ENTROPY METHOD WITH MONOTONIC NONLINEAR CRITIC DECOMPOSITION

## ABSTRACT

Cooperative multi-agent reinforcement learning (MARL) commonly adopts centralized training with decentralized execution (CTDE), where centralized critics leverage global information to guide decentralized actors. However, centralized-decentralized mismatch (CDM) arises when the suboptimal behavior of one agent degrades others' learning. Prior approaches mitigate CDM through value decomposition, but linear decompositions allow per-agent gradients at the cost of limited expressiveness, while nonlinear decompositions improve representation but require centralized gradients, reintroducing CDM. To overcome this trade-off, we propose the multi-agent cross-entropy method (MCEM), combined with monotonic nonlinear critic decomposition (NCD). MCEM updates policies by increasing the probability of high-value joint actions, thereby excluding suboptimal behaviors. For sample efficiency, we extend off-policy learning with a modified $k$-step return and Retrace. Analysis and experiments demonstrate that MCEM outperforms state-of-the-art methods across both continuous and discrete action benchmarks.

## 1 INTRODUCTION

Cooperative multi-agent reinforcement learning (MARL) has made significant progress in recent years, see (Oroojlooy & Hajinezhad, 2023) for a comprehensive review. A widely adopted paradigm is centralized training with decentralized execution (CTDE), which underpins both value-based and policy gradient approaches. Prominent policy gradient approaches, such as COMA (Foerster et al., 2018), MADDPG (Lowe et al., 2017), and MAPPG (Chen et al., 2023), follow the centralized critic with decentralized actors (CCDA) framework. Each agent has a centralized critic that leverages full information—including the global state and the information of other agents—to evaluate the agent's behaviors. However, a key limitation of this setup is that the suboptimal behavior of one agent can influence the centralized critics of other agents and, in turn, degrade their learning, known as *centralized-decentralized mismatch* (CDM) (Wang et al., 2020b).

To address CDM, Wang et al. (2020b) propose DOP that represents the centralized critic as a weighted sum of individual agent critics. The linear decomposition allows for per-agent gradients when updating agents' policies. The suboptimal behavior of one agent is isolated and does not degrade the policy learning of other agents. However, the linear decomposition has limited representation capability. The centralized critic may be misallocated to individual agent critics and therefore negatively affect the per-agent gradient. Peng et al. (2021) propose FACMAC to factor the centralized critic across agents with a flexible decomposition function. For the same, (Su et al., 2021) proposes using a linear decomposition function (VDAC-sum) and a monotonic decomposition function (VDAC-mix), and LICA (Zhou et al., 2020) uses a nonlinear function. While enjoying rich expressiveness, the nonlinear decomposition must use a centralized gradient when updating the policies of agents. As a result, when one agent exhibits suboptimal behavior, it can adversely affect the learning of other agents, indicating that CDM remains unresolved.

To overcome the limitations of existing methods, this study proposes a new method MCEM-NCD (Multi-agent Cross-Entropy Method with Monotonic Nonlinear Critic Decomposition), which can successfully address the CDM issue and enjoy the rich representation capability of the nonlinear critic decomposition. Our method is characterized by learning the agent policies with an extended Cross-Entropy Method (CEM) (Rubinstein, 1999). CEM is a sampling-based stochastic optimization technique and has been adopted in reinforcement learning domains (Kalashnikov et al., 2018; Simmons-Edler et al., 2019; Shao et al., 2022; Neumann et al., 2023). In this study, we extend the CEM framework from single-agent to multi-agent CEM (MCEM). Specifically, a batch of joint

actions is sampled following current policies of agents and only the joint actions with the best performance contribute to policy learning. By this, the joint actions of poor performance due to the suboptimal behavior of a few agents are excluded from the policy learning and thus effectively address CDM. A monotonic nonlinear decomposition function is applied to factor the centralized critic into individual agent critics. The monotonicity ensures that the best performance of the individual agents aligns with the best performance of the agents as a team. MCEM-NCD supports stochastic policies with both discrete actions and continuous actions.

For sample efficiency when learning the centralized but factored critic, we employ the $k$-step $\lambda$-return off-policy TB (Tree-backup) method (Wang et al., 2020b). This method is based on Expected Sarsa (Sutton & Barto, 2018), which requires the expected value over all actions and agents, and thus suffers from intractable computational complexity. Wang et al. (2020b) shows that the linear decomposition can significantly reduce the computation. To work with nonlinear decomposition, we replace the Expected Sarsa with Sarsa. Moreover, we optimize off-policy learning by introducing Retrace method (Munos et al., 2016) to overcome the limitation of TB. The analysis and experiments demonstrate the superiority of MCEM-NCD, showing strong performance with discrete actions on various StarCraft Multi-Agent Challenge (SMAC) scenarios, especially the *Hard* and *Super Hard* ones (Samvelyan et al., 2019), as well as with continuous actions in Continuous Predator-Prey environments (Peng et al., 2021). To facilitate reproducibility, the code for this study is available at `https://github.com/Yuru27/MCEM-NCD`.

## 2 RELATED WORK

A widely adopted MARL paradigm is centralized training with decentralized execution (CTDE), including value-based approaches (Rashid et al., 2020a;b; Son et al., 2019; Sunehag et al., 2018; Wang et al., 2020a) and policy gradient approaches (Foerster et al., 2018; Kuba et al., 2021; Lowe et al., 2017; Yu et al., 2022; Wang et al., 2020b; Peng et al., 2021). The value-based approaches aim to decompose the joint action-value across individual agents where the Individual-Global-Max (IGM) principle is widely applied, i.e., the global optimal action should align with the collection of individual optimal actions of agents (Rashid et al., 2020b). Following IGM, Sunehag et al. (2018) propose value decomposition networks (VDN) to represent the joint action-value function as a sum of value functions of individual agents. QMIX changes the simple sum function of the VDN to a deep neural network that satisfies the monotonicity constraints (Rashid et al., 2020b). Various value-based approaches have been developed to enhance the expressive power of QMIX, including Qatten (Yang et al., 2020), QPLEX (Wang et al., 2020a), QTRAN (Son et al., 2019; 2020), MAVEN (Mahajan et al., 2019), WQMIX (Rashid et al., 2020a), TLMIX (Zhang et al., 2023), DFAC (Sun et al., 2021), and SMIX (Wen et al., 2020). The value-based approaches are mainly limited to discrete actions.

When facing tasks with continuous actions, multi-agent policy gradient methods are often considered. Some actor-critic approaches adopt a centralized critic with the decentralized actors framework (CCDA) (Lowe et al., 2017; Foerster et al., 2018; Chen et al., 2023). By extending its single-agent counterpart (DDPG) (Lillicrap et al., 2016), MADDPG (Lowe et al., 2017) learns policies by approximating a centralized critic for each agent, which requires knowledge of the policies of other agents. COMA (Foerster et al., 2018) learns a centralized critic function $Q_{tot}$ and then each agent models an advantage function for its current action by comparing $Q_{tot}$ with a counterfactual baseline, where a default action replaces the current action of the agent, but the other agents maintain their actions. The advantage function is used to update the policy of the agent. MAPPG (Chen et al., 2023) learns a centralized critic for each agent via a polarization policy gradient. The methods under CCDA suffer from the issue of *centralized-decentralized mismatch* (CDM) (Wang et al., 2020b), that is, the suboptimality of one agent can propagate through the centralized critic and negatively affect policy learning of other agents, causing catastrophic miscoordination.

Wang et al. (2020b) address CDM by proposing DOP that factors the centralized critic into individual agent critics using a linear decomposition function. The linear decomposition has limited representational capability, but it is essential for policy update based on per-agent gradients. The suboptimal behavior of one agent is isolated and does not degrade the policy learning of other agents. FACMAC (Peng et al., 2021) extends the MADDPG framework by introducing a centralized, yet factored, critic with a neural network like QMIX, but relaxes its original monotonicity constraints to enable more flexible decomposition of value functions. VDAC (Su et al., 2021) proposes using the monotonic decomposition function to factor the centralized critic into individual agent critics, including a linear decomposition function (VDAC-sum) and a monotonic decomposition function (VDAC-mix). For

both FACMAC and VDAC, the policy update is based on the centralized gradient if the decomposition function is nonlinear; otherwise, it is based on the per-agent gradients. LICA (Zhou et al., 2020) factors the centralized critic across agents by training a nonlinear function to establish the relationship between the joint action-value and the action samples drawn from policies of individual agents. For both LICA and VDAC, the policies of agents are updated based on the centralized gradient. These studies reveal a trade-off: linear decompositions enable per-agent gradients to mitigate CDM but lack expressiveness, whereas nonlinear decompositions enhance representation yet rely on centralized gradients, reintroducing the issue of CDM. This work aims to address this problem.

Zhang et al. (2021) introduces FOP and shows that it enables agents to perform globally optimal behavior. However, its underlying assumption—that the optimal joint action-value function can be factored into individual agent functions—is often too restrictive, failing to capture necessary joint behavior in tasks requiring significant coordination (Cassano & Sayed, 2021). Other actor-critic studies investigate collaborative MARL from specific perspectives and are orthogonal to our study. (Zhou et al., 2022) investigates the situation that one agent may have the same partial observations at different states. (Zang et al., 2023) evaluates agents in a sequence, i.e., the less-affected agents are evaluated before other agents.

## 3 BACKGROUND

### 3.1 MODEL

We take DEC-POMDP (Oliehoek et al., 2016) for modeling cooperative multi-agent tasks as a tuple $< A, S, U, Z, P, R, k, \gamma >$ where $A = \{a_1, \ldots, a_k\}$ represents $k$ agents, $s \in S$ denotes the true state of the environment, $U$ is the action space, and $\gamma \in [0, 1)$ is the discount factor. We consider a partially observable scenario in which each agent draws individual observations $z \in Z$ according to an observation function $O(s, a) : S \times A \mapsto Z$.

At each time step, each agent $a \in A$ has an action-observation history $\tau^a \in T \equiv (Z \times U)^*$. The agent follows a stochastic policy $\pi^a : T \times U \to [0, 1]$, where $\pi^a(u^a \mid \tau^a)$ denotes the probability of choosing action $u^a \in U$ given history $\tau^a$. All agents collectively define a joint policy $\boldsymbol{\pi} = \{\pi^a\}_{a \in A}$ and a joint action–observation history $\boldsymbol{\tau} = \{\tau^a\}_{a \in A}$. The actions chosen by all agents together form the joint action $\boldsymbol{u} = \{u^a\}_{a \in A}$. The joint action space is denoted as $\boldsymbol{U}$. Executing a joint action $\boldsymbol{u}$ causes the transition of the environment from state $s \in S$ to state $s' \in S$ according to the transition function $P(s'|s, \boldsymbol{u}) : S \times \boldsymbol{U} \times S \mapsto [0, 1]$. All agents share the same reward function $r(\boldsymbol{\tau}, \boldsymbol{u}) : \boldsymbol{U} \times (Z \times U)^{*|A|} \mapsto \mathbb{R}$. For the joint policy $\boldsymbol{\pi}$, the joint action-value function is $Q_{tot}(\boldsymbol{\tau}_t, \boldsymbol{u}_t) = \mathbb{E}_{\boldsymbol{\tau}_{t+1:\infty}, \boldsymbol{u}_{t+1:\infty}}[R_t|\boldsymbol{\tau}_t, \boldsymbol{u}_t]$, where $R_t = \sum_{i=0}^{\infty} \gamma^i r_{t+i}$ is the discounted return.

We consider both discrete and continuous action spaces, for which stochastic policies are learned. Although training is centralized, execution is decentralized. That is, the learning algorithm has access to all local action-observation histories $\boldsymbol{\tau}$ and global state $s$, but each agent $a$ learns policy conditional only on its own action-observation history $\tau^a$.

### 3.2 CHALLENGES

DOP (Wang et al., 2020b) factors the centralized critic into individual agent critics using a linear decomposition function:

$$Q_{tot}(\boldsymbol{\tau}, \boldsymbol{u}) = \sum_{a \in A} w^a(\boldsymbol{\tau}) Q^a(\boldsymbol{\tau}, u^a; \phi^a) + b(\boldsymbol{\tau}) \tag{1}$$

where $\phi^a$ are parameters of local action-value function $Q^a$ of agent $a$; $w^a(\boldsymbol{\tau}) > 0$ (the weight associated with agent $a$) and $b(\boldsymbol{\tau})$ are generated by the learnable networks. Similarly, the linear decomposition function can be applied in FACMAC (Peng et al., 2021) and VDAC (Su et al., 2021). With the linearly decomposed critic architecture, learning stochastic policies with discrete actions is based on the per-agent policy gradients defined as:

$$\nabla J(\boldsymbol{\theta}) = \mathbb{E}_{\substack{\boldsymbol{u} \sim \boldsymbol{\pi} \\ \boldsymbol{\tau} \sim \mathcal{D}}} \left[ \sum_{a \in A} w^a(\boldsymbol{\tau}) \nabla_{\theta^a} \log \pi^a(u^a|\tau^a; \theta^a) Q^a(\boldsymbol{\tau}, u^a; \phi^a) \right] \tag{2}$$

where the actor network of agent $a$ is parameterized by $\theta^a$ and the parameters of all agents together are denoted $\boldsymbol{\theta}$. Under the per-agent policy gradients, the policy of agent $a$ is updated with respect to $Q^a(\boldsymbol{\tau}, u^a; \phi^a)$, independent from the local action-values of other agents. Consequently, the suboptimal behavior of one agent does not propagate into the policy gradient updates of others. However, due to the limited representational capacity of linear functions, $Q_{tot}(\boldsymbol{\tau}, \boldsymbol{u})$ may be improperly allocated to $Q^a(\boldsymbol{\tau}, u^a; \phi^a)$ in complicated scenarios, thereby impairing policy learning.

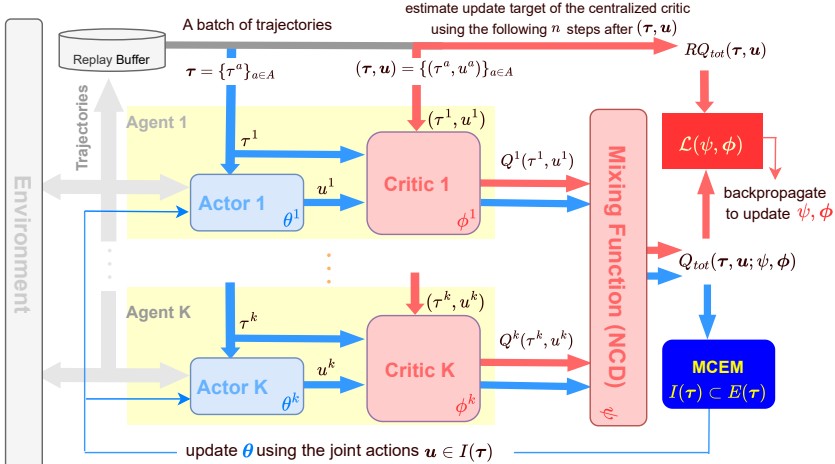

Figure 1: The process flow of decentralized policy learning with multi-agent CEM (MCEM) is marked in blue (detailed in section 4.1). The process flow of the off-policy method for learning a centralized but factored critic is marked in red (detailed in section 4.2).

The nonlinear decomposition function in FACMAC (Peng et al., 2021) and VDAC-mix (Su et al., 2021) can be applied to factor the centralized critic into individual agent critics:

$$Q_{tot}(\boldsymbol{\tau}, \boldsymbol{u}) = F(Q^{a_1}(\tau^{a_1}, u^{a_1}), \cdots, Q^{a_k}(\tau^{a_k}, u^{a_k}); \psi) \tag{3}$$

where the nonlinear function $F$ is parameterized by $\psi$. These methods rely on the centralized gradient to learn agent policies. For stochastic policies with discrete actions, the centralized policy gradient is given by

$$\nabla J(\boldsymbol{\theta}) = \mathbb{E}_{\substack{\boldsymbol{u} \sim \boldsymbol{\pi} \\ \boldsymbol{\tau} \sim \mathcal{D}}} \left[ \sum_{a \in A} \nabla_{\theta^a} \log \pi^a(u^a | \tau^a; \theta^a) Q_{tot}(\boldsymbol{\tau}, \boldsymbol{u}) \right] \tag{4}$$

Due to the centralized policy gradient, the suboptimal behavior of a single agent can reduce the joint action–value $Q_{tot}$. As a result, other agents—despite taking optimal actions—are forced to update their policies in an unfavorable direction. This misattribution of credit is precisely the CDM problem.

## 4 METHOD

We propose MCEM-NCD (Multi-agent Cross-Entropy Method with Monotonic Nonlinear Critic Decomposition). The overall framework of MCEM-NCD is illustrated in fig. 1, and the corresponding pseudocode is provided in algorithm 1 (appendix C). In practice, NCD is a monotonic nonlinear network, parameterized by $\psi$, following QMIX (Rashid et al., 2020b). The inputs are the action-values produced by the individual agent critics, $\{Q^a\}_{a \in A}$, and the output is the joint action-value $Q_{tot}$. It ensures that the global $\arg\max$ performed on $Q_{tot}$ yields the same result as a set of individual $\arg\max$ operations performed on each $Q^a$:

$$\arg\max_{\boldsymbol{u}} Q_{tot}(\boldsymbol{\tau}, \boldsymbol{u}) = \{\arg\max_{u^a} Q^a(\tau^a, u^a)\}_{a \in A} \tag{5}$$

By this, each agent $a$ participates in a decentralized execution solely by choosing greedy actions concerning its $Q^a$. Monotonicity can be enforced by restricting the relationship between $Q_{tot}$ and each $Q^a$: $\frac{\partial Q_{tot}}{\partial Q^a} \geq 0$ for $a \in A$.

To improve sample efficiency, MCEM-NCD leverages off-policy data to train the policy networks (actors), action-value functions (critics), and the monotonic nonlinear critic decomposition network. During training, agents periodically sample actions from their current policies, and the resulting interactions with the environment generate episodes that are stored in a replay buffer $\mathcal{D}$. Each episode record at time step $t$ is represented as $(s_t, \boldsymbol{\tau}_t, \boldsymbol{u}_t, r_t, s_{t+1})$.

### 4.1 DECENTRALIZED POLICY LEARNING WITH MULTI-AGENT CEM

The cross-entropy method (CEM) (Rubinstein, 1999) is a sampling-based stochastic optimization technique that employs iterative updates to the sampling distribution. This methodology has seen

increasing adoption in reinforcement learning domains (Kalashnikov et al., 2018; Simmons-Edler et al., 2019; Shao et al., 2022; Neumann et al., 2023) and is categorized as evolutionary reinforcement learning (Lin et al., 2025). Existing studies focus on single-agent reinforcement learning. We extend the single-agent CEM framework in (Neumann et al., 2023) to the multi-agent CEM (MCEM). MCEM randomly samples records from the replay buffer $\mathcal{D}$ and, for each record, takes the following phases:

1. *Sampling:* For $\boldsymbol{\tau} = \{\tau^a\}_{a \in A}$ in each record, we draw a set of joint actions $E(\boldsymbol{\tau})$ from the decentralized policies $\{\pi^a(\cdot|\tau^a; \theta^a)\}_{a \in A}$ which are parameterized by $\{\theta^a\}_{a \in A}$.

2. *Evaluation:* For each joint action $\boldsymbol{u} \in E(\boldsymbol{\tau})$, we compute $Q_{tot}(\boldsymbol{\tau}, \boldsymbol{u})$.

3. *Elite Selection:* We select a subset of joint actions $I(\boldsymbol{\tau}) \subset E(\boldsymbol{\tau})$ where $Q_{tot}(\boldsymbol{\tau}, \boldsymbol{u})$ for $\boldsymbol{u} \in I(\boldsymbol{\tau})$ is in the top $(1 - \rho)$ quantile values.

For each record, we repeat the three phases and use $\{I(\boldsymbol{\tau})\}_{\tau \sim \mathcal{D}}$ to update decentralized actors (policies). To help agents learn their policies $\boldsymbol{\pi} = \{\pi^a\}_{a \in A}$, called *main policies*, we adopt auxiliary *proposal policies* $\hat{\boldsymbol{\pi}} = \{\hat{\pi}^a\}_{a \in A}$, one for each agent, following (Neumann et al., 2023). These proposal policies are entropy regularized to ensure that we keep a broader set of potential actions in the sampling phase. The main policies do not use entropy regularization, allowing them to start acting according to currently greedy actions more quickly. Updating the main policies and the proposal policies are based on the gradient in eq. (6) and eq. (7), respectively:

$$\nabla J(\boldsymbol{\theta}) = \mathbb{E}_{\boldsymbol{\tau} \sim \mathcal{D}} \left[ \sum_{\boldsymbol{u} \in I(\boldsymbol{\tau})} \sum_{a \in A} \nabla_{\theta^a} \log \pi^a(u^a | \tau^a; \theta^a) \right] \tag{6}$$

$$\nabla J(\hat{\boldsymbol{\theta}}) = \mathbb{E}_{\boldsymbol{\tau} \sim \mathcal{D}} \left[ \sum_{\boldsymbol{u} \in I(\boldsymbol{\tau})} \sum_{a \in A} \nabla_{\hat{\theta}^a} \log \hat{\pi}^a(u^a | \tau^a; \hat{\theta}^a) + \beta \nabla_{\hat{\theta}^a} \mathcal{H}(\hat{\pi}^a(\cdot | \tau^a; \hat{\theta}^a)) \right] \tag{7}$$

where $\mathcal{H}(\hat{\pi}^a(\cdot | \tau^a; \hat{\theta}^a))$ is the entropy regularizer, the main policy $\pi^a$ and the proposal policy $\hat{\pi}^a$ are parameterized by $\theta^a$ and $\hat{\theta}^a$, respectively, for agent $a$.

**Stochastic Policies with Continuous Actions** Learning stochastic policies in discrete action spaces is well supported by MCEM as discussed above. For RL with continuous actions, the policy gradient methods are often considered and can be divided into two main categories: deterministic and stochastic. Deterministic policies are often more sample-efficient than stochastic policies due to lower gradient variance (Silver et al., 2014; Schulman et al., 2015). However, they lack inherent exploration and require external noise (Lillicrap et al., 2016). In contrast, stochastic policies possess built-in exploration (Williams, 1992; Sutton & Barto, 2018). This inherent exploratory capability not only makes stochastic policies more robust in POMDPs where state is a probability distribution over the true states (Singh et al., 1994; Dutech & Scherrer, 2013), but also enables them to potentially avoid local optima and discover better final policies (Haarnoja et al., 2018; Ahmed et al., 2019).

For MARL with continuous actions, many existing policy gradient methods learn deterministic policies, including MADDPG (Lowe et al., 2017), DOP (Wang et al., 2020b), and FACMAC (Peng et al., 2021). Some studies facilitate learning stochastic policies where the stochastic policies are parameterized as Gaussian distributions over continuous actions. Foerster et al. (2018) mentioned that COMA can be readily extended to continuous action domains by employing Gaussian policies. MAPPG (Chen et al., 2023) and FOP (Zhang et al., 2021) utilize Gaussian distributions for action sampling in their continuous control settings.

MCEM inherently supports stochastic policies in a continuous action space. Specifically, the built-in exploratory nature of stochastic policies enables the *sampling* phase of MCEM. For each agent $a$, the stochastic policy is a Gaussian distribution over continuous action:

$$\pi^a(u^a | \tau^a; \theta^a) = \frac{1}{\sigma^a \sqrt{2\pi}} \exp\left(-\frac{(u^a - \mu^a)^2}{2(\sigma^a)^2}\right) \tag{8}$$

where the variance $\sigma^a$ and mean $\mu^a$ of the Gaussian distribution are returned by a network parameterized by $\theta^a$ for input $\tau^a$. The formula for the entropy regularization of the Gaussian distribution is as follows:

$$\mathcal{H}(\pi^a(\cdot | \tau^a; \theta^a)) = \frac{1}{2} \log(2\pi e(\sigma^a)^2) \tag{9}$$

By replacing $\pi^a(u^a | \tau^a; \theta^a)$ in eq. (6) by eq. (8), we can update the main policies in continuous action space. By replacing $\mathcal{H}(\hat{\pi}^a(\cdot | \tau^a; \theta^a))$ in eq. (7) by eq. (9) and reformulating $\hat{\pi}^a(u^a | \tau^a; \hat{\theta}^a)$ in eq. (7) as eq. (8), we can update the proposal policies in a continuous action space. More details of stochastic polices with continuous actions are presented in appendix B

### 4.2 OFF-POLICY METHOD FOR LEARNING CENTRALIZED BUT FACTORED CRITIC

For sample efficiency when estimating the update target of the centralized critic, we explore off-policy data in the replay buffer, that is, the episodes generated following behavior policy $\beta$. To this end, we adapt the following operator (Wang et al., 2020b):

$$\mathcal{R}Q_{tot}(\boldsymbol{\tau}, \boldsymbol{u}) = Q_{tot}(\boldsymbol{\tau}, \boldsymbol{u}) + \mathbb{E}_{\boldsymbol{\beta}} \left( \sum_{t \geq 0}^{n-1} \gamma^t (\prod_{j=1}^t c_j) \delta_t \right) \tag{10}$$

In (Wang et al., 2020b), $\delta_t = r_t + \gamma \mathbb{E}_{\boldsymbol{\pi}}[Q_{tot}(\boldsymbol{\tau}_{t+1}, \cdot)] - Q_{tot}(\boldsymbol{\tau}_t, \boldsymbol{u}_t)$ and the nonnegative coefficient $c_j = \lambda \boldsymbol{\pi}(\boldsymbol{u}_j|\boldsymbol{\tau}_j) = \lambda \prod_{a \in A} \pi^a(u_j^a|\tau_j^a)$. It is the multi-agent version of the $n$-step $\lambda$-return off-policy TB algorithm (Munos et al., 2016) in the Expected Sarsa form (Sutton & Barto, 2018). However, such definitions of $\delta_t$ and $c_j$ have limitations.

The definition of $\delta_t$ requires $\mathbb{E}_{\boldsymbol{\pi}}[Q_{tot}(\boldsymbol{\tau}_{t+1}, \cdot)] = \sum_{\boldsymbol{u} \in U} \pi(\boldsymbol{u}|\boldsymbol{\tau}_{t+1})Q(\boldsymbol{\tau}_{t+1}, \boldsymbol{u})$ for discrete actions, which needs $O(|U|^k)$ steps of summation. With the linearly decomposed critic in (Wang et al., 2020b), the complexity of computing $\mathbb{E}_{\boldsymbol{\pi}}[Q_{tot}(\boldsymbol{\tau}_{t+1}, \cdot)]$ is reduced to $O(n|U|)$. Since our method uses a nonlinear decomposition function to factorize the centralized critic, we adopt:

$$\delta_t = r_t + \gamma Q_{tot}(\boldsymbol{\tau}_{t+1}, \boldsymbol{u}_{t+1}) - Q_{tot}(\boldsymbol{\tau}_t, \boldsymbol{u}_t) \tag{11}$$

where $Q_{tot}(\boldsymbol{\tau}_{t+1}, \boldsymbol{u}_{t+1})$ features the $\lambda$-return off-policy in the Sarsa form (Sutton & Barto, 2018). That is, $\boldsymbol{u}_{t+1}$ is the next joint action chosen following the current policy $\boldsymbol{\pi}(\cdot|\boldsymbol{\tau}_{t+1})$ given $\boldsymbol{\tau}_{t+1}$, i.e., $\boldsymbol{u}_{t+1} \sim \boldsymbol{\pi}(\cdot|\boldsymbol{\tau}_{t+1})$. More interestingly, it is ready to support continuous actions. In contrast, it is hard, if not impossible, for $\delta_t$ defined in (Wang et al., 2020b) to work with continuous actions, even though the linearly decomposed critic is applied.

The nonnegative coefficient $c_j = \lambda \boldsymbol{\pi}(\boldsymbol{u}_j|\boldsymbol{\tau}_j) = \lambda \prod_{a \in A} \pi^a(u_j^a|\tau_j^a)$ corrects the discrepancy between $\boldsymbol{\pi}$ and $\boldsymbol{\beta}$ when learning from the off-policy returns following behavior policy $\boldsymbol{\beta}$. But it is inefficient in the case of near policy where $\boldsymbol{\beta}$ and $\boldsymbol{\pi}$ are similar (Munos et al., 2016). Another popular method is Importance Sampling (IS) $c_j = \frac{\boldsymbol{\pi}(\boldsymbol{a}_j|\boldsymbol{\tau}_j)}{\boldsymbol{\beta}(\boldsymbol{a}_j|\boldsymbol{\tau}_j)}$. Due to the product $\prod_{j=1}^t c_j$, IS suffers from large, even possibly infinite, variance. To address the limitations, we adopt the Retrace algorithm $c_j = \lambda \min(1, \frac{\boldsymbol{\pi}(\boldsymbol{a}_j|\boldsymbol{\tau}_j)}{\boldsymbol{\beta}(\boldsymbol{a}_j|\boldsymbol{\tau}_j)}$ (Munos et al., 2016) which incorporates a correction mechanism to IS. By truncating the importance weight at 1, Retrace effectively reduces the variance of IS and therefore improves the stability of the off-policy action-value estimators.

During learning, as illustrated in fig. 1, we randomly sample a trajectory from the replay buffer $\mathcal{D}$ and extract $(\boldsymbol{\tau}, \boldsymbol{u})$ from each record in the trajectory. For agent $a$, we estimate the local critic $Q^a(\tau^a, u^a; \phi^a)$ and then use these local values as inputs to compute the joint action-value $Q_{tot}(\boldsymbol{\tau}, \boldsymbol{u}; \psi, \boldsymbol{\phi})$ through the mixing function NCD, parameterized by $\psi$. From the same trajectory, the following $n$ steps after the trajectory record, where $(\boldsymbol{\tau}, \boldsymbol{u})$ is extracted, are explored to return $\mathcal{R}Q_{tot}(\boldsymbol{\tau}, \boldsymbol{u})$ using operator in eq. (10). $\mathcal{R}Q_{tot}(\boldsymbol{\tau}, \boldsymbol{u})$ is the update target of $Q_{tot}(\boldsymbol{\tau}, \boldsymbol{u}; \boldsymbol{\phi}, \psi)$. The parameters of local critics, i.e., $\boldsymbol{\phi} = \{\phi^a\}_{a \in A}$, and the parameters of the mixing function, $\psi$, are updated by minimizing the loss:

$$\mathcal{L}(\psi, \boldsymbol{\phi}) = \mathbb{E}_{(\boldsymbol{\tau}, \boldsymbol{u}) \sim \mathcal{D}} \left[ (\mathcal{R}Q_{tot}(\boldsymbol{\tau}, \boldsymbol{u}) - Q_{tot}(\boldsymbol{\tau}, \boldsymbol{u}; \psi, \boldsymbol{\phi}))^2 \right] \tag{12}$$

$$Q_{tot}(\boldsymbol{\tau}, \boldsymbol{u}; \psi, \boldsymbol{\phi}) = \text{NCD} \left( Q^1(\tau^1, u^1; \phi^1), \cdots, Q^k(\tau^k, u^k; \phi^k), \psi \right) \tag{13}$$

## 5 ANALYSIS

To distinguish, we refer to the agent policies learned using MCEM-NCD as *percentile-greedy policies* $\boldsymbol{\pi}_\rho = \{\pi_\rho^a\}_{a \in A}$, which updates agent policies by only raising the probability of joint actions in the top $(1 - \rho)$ quantile according to $Q_{tot}(\boldsymbol{\tau}, \cdot)$; and refer to the agent policies learned with the centralized gradient (eq. (4)) as *centralized gradient policies* $\boldsymbol{\pi}_g = \{\pi_g^a\}_{a \in A}$.

**Theorem 5.1.** *The percentile-greedy policy $\boldsymbol{\pi}_\rho$, where $\rho > 0$, is guaranteed to be at least as good as the centralized gradient policies $\boldsymbol{\pi}_g$ for any given $\boldsymbol{\tau}$. It can be formulated as eq. (14) for discrete actions:*

$$\sum_{\boldsymbol{u} \in U} \boldsymbol{\pi}_\rho(\boldsymbol{u}|\boldsymbol{\tau})Q_{tot}^{\boldsymbol{\pi}_\rho}(\boldsymbol{\tau}, \boldsymbol{u}) \geq \sum_{\boldsymbol{u} \in U} \boldsymbol{\pi}_g(\boldsymbol{u}|\boldsymbol{\tau})Q_{tot}^{\boldsymbol{\pi}_g}(\boldsymbol{\tau}, \boldsymbol{u}) \tag{14}$$

*For continuous actions, it can be formulated as eq. (15):*

$$\int_U \boldsymbol{\pi}_\rho(\boldsymbol{u}|\boldsymbol{\tau})Q_{tot}^{\boldsymbol{\pi}_\rho}(\boldsymbol{\tau}, \boldsymbol{u})d\boldsymbol{u} \geq \int_U \boldsymbol{\pi}_g(\boldsymbol{u}|\boldsymbol{\tau})Q_{tot}^{\boldsymbol{\pi}_g}(\boldsymbol{\tau}, \boldsymbol{u})d\boldsymbol{u} \tag{15}$$

The proof is in appendix A

# 6 EXPERIMENTS

This section presents our experimental results on the discrete-action SMAC benchmark (Samvelyan et al., 2019) and the continuous-action Predator-Prey benchmark (Peng et al., 2021). The experiments were conducted on a workstation equipped with a 32-Core AMD Ryzen Threadripper PRO 7975WX processor, 128 GB of RAM, and an NVIDIA RTX 4080 GPU (16 GB VRAM). The evaluation metric is the *median win rate* for discrete tasks and the *mean episode return* for continuous tasks when the number of training steps increases. To ensure reliable results, we repeat every experiment by three times with different random seeds. For our MCEM-NCD, the default setting of percentile parameter $\rho = 0.9$ for continuous tasks, $\rho = 0.8$ for discrete tasks; the default number of joint actions sampled per $\tau$ (i.e., the size of $E(\tau)$ in section 4.1) is 20 for continuous tasks, 10 for discrete tasks. For baseline methods, we employ the hyperparameter settings from the source of the code.

## 6.1 DISCRETE ACTION TASKS

StarCraft II (SC2) contains a variety of specialized units, each possessing unique capabilities that enable the development of complex cooperative strategies among agents. The StarCraft Multi-Agent Challenge (SMAC) (Samvelyan et al., 2019) comprises a set of StarCraft II micromanagement battle scenarios designed to evaluate how independent agents can cooperate to solve complex tasks. This study evaluates the performance of the proposed MCEM-NCD on SC2 micromanagement tasks in the SMAC benchmark. These tasks require decentralized agents to control a group of heterogeneous ally units

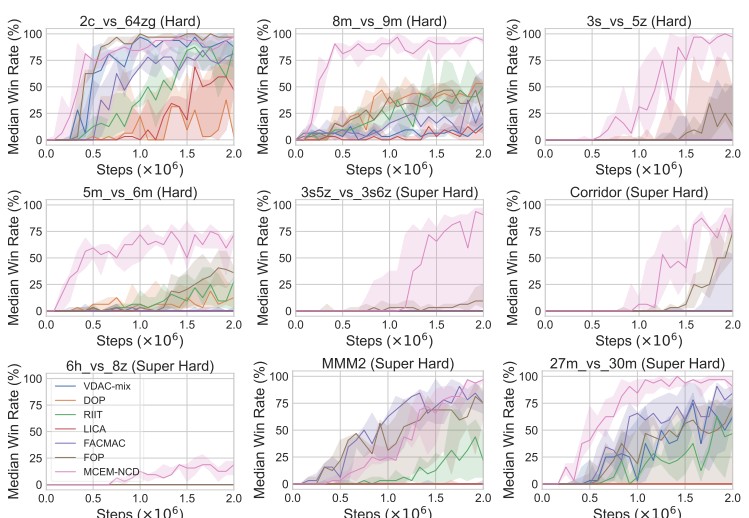

Figure 2: Discrete action tasks - performance measured by *median win rate* on 9 scenarios in the SMAC benchmark.

and collaboratively defeat a comparable heterogeneous group of enemy units governed by the built-in game AI. Battles may occur in either symmetric scenarios, where both groups have identical units, or asymmetric scenarios, featuring divergent units.

The default environment settings in SMAC are employed. Each episode terminates under two conditions: either when all units from one faction have been eliminated, or when the predefined step limit for the episode is reached. A win is recorded when all enemy units are successfully defeated. The game framework categorizes the scenarios into three difficulty levels: *Easy*, *Hard*, and *Super Hard*. In our experiments, we examined 9 battle scenarios, including 4 Hard (2c_vs_64zg, 8m_vs_9m, 3s_vs_5z, and 5m_vs_6m) and 5 Super Hard (3s5z_vs_3s6z, corridor, 6h_vs_8z, MMM2, and 27m_vs_30m). We compared our MCEM-NCD on these scenarios against six state-of-the-art baselines. For LICA (Zhou et al., 2020), RIIT (Hu et al., 2021), VDAC-mix (Su et al., 2021), and DOP (Wang et al., 2020b), the code is provided by PyMARL2 [1] (Hu et al., 2021). For FACMAC (Peng et al., 2021) and FOP (Zhang et al., 2021), the code is provided by the authors [2].

As shown in fig. 2, MCEM-NCD achieves outstanding performance in terms of median win rate and convergence speed across all nine scenarios. In the 2c_vs_64zg scenario, many methods perform

---

[1] https://github.com/hijkzzz/pymarl2.

[2] https://github.com/oxwhirl/facmac (**FACMAC**), https://github.com/liyheng/FOP (**FOP**)

strongly due to their ability to control two Colossus units optimally. MCEM-NCD matches the two best baselines and delivers significantly better performance than other baselines. Additionally, MCEM-NCD exhibits a small variance in the median win rate. In the 3s_vs_5z scenario, where allied agents must kite enemies over long episodes (lasting at least 100 timesteps), i.e., rewards are substantially delayed (Samvelyan et al., 2019), MCEM-NCD achieves the best results, demonstrating both superior median win rate and the fastest convergence, with consistently lower variance than the baselines. In asymmetric matchups such as 8m_vs_9m, 5m_vs_6m, and 27m_vs_30m, success depends on precise coordination and extensive exploration to discover advanced collaborative tactics. Here, MCEM-NCD again dominates in both median win rate and convergence speed. Finally, in the most challenging heterogeneous and asymmetric settings–3s5z_vs_3s6z, corridor, 6h_vs_8z, and MMM2–baselines struggle to achieve meaningful performance, whereas MCEM-NCD consistently converges to significantly higher median win rates.

## 6.2 CONTINUOUS ACTION TASKS

For continuous action tasks, we evaluate MCEM-NCD on the Continuous Predator-Prey environment (Peng et al., 2021), a continuous-action variant of the classic predator-prey game. In this setting, $N$ slower cooperative agents pursue $M$ faster prey on a two-dimensional plane with two large, randomly placed landmarks

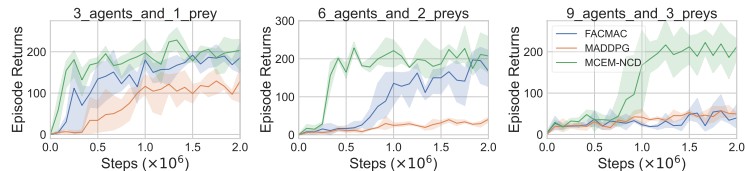

Figure 3: Continuous action tasks - performance measured by *mean episode return* on 3 scenarios in continuous Predator-Prey with varying numbers of agents and prey.

serving as obstacles. Each agent $a$ selects a movement action $u^a \in \mathbb{R}^2$. A cooperative capture, where one agent collides with a prey while others are within a specified proximity, yields a team reward of +10. In contrast, an isolated capture incurs a penalty of -1, and all other outcomes yield 0. The environment is partially observable, as each agent perceives only entities (agents, prey, and landmarks) within its limited view radius. We test our method on three scenarios: 3_agents_and_1_prey, 6_agents_and_2_preys, and 9_agents_and_3_preys.

With continuous actions, MADDPG (Lowe et al., 2017), DOP (Wang et al., 2020b), and FACMAC (Peng et al., 2021) adopt deterministic policies, but only MADDPG and FACMAC provide publicly available implementations[3]. In contrast, COMA (Foerster et al., 2018), MAPPG (Chen et al., 2023), and FOP (Zhang et al., 2021) employ stochastic policies, yet none of them release source code. To ensure fair and reproducible

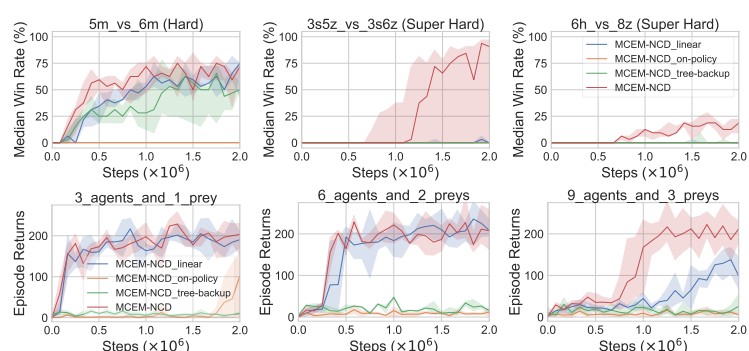

Figure 4: Ablation study on the discrete tasks (top 3) and on the continuous tasks (bottom 3).

comparisons, we restrict our experiments to algorithms with publicly available and previously validated implementations, thereby avoiding unfair results caused by potential reimplementation bias.

The experimental results are presented in fig. 3. Compared with the baselines, our MCEM-NCD achieves substantially better performance in terms of both *mean episode return* and convergence speed across all scenarios, while all methods show a comparable level of variance. As expected, the performance of all methods decreases as task complexity increases, i.e., with higher mean episode return in simpler settings (involving fewer agents and prey) and lower mean episode return in more

---

[3]https://github.com/oxwhirl/facmac

challenging ones. Notably, the relative advantage of MCEM-NCD becomes more pronounced in complex environments, indicating that our approach not only scales effectively but also remains robust when coordination demands intensify. These results underscore the strength of MCEM-NCD in addressing increasingly difficult multi-agent interactions, where conventional baselines struggle.

### 6.3 ABLATION STUDY

*Linear vs Nonlinear Critic Decomposition*: Our MCEM-NCD is featured by multi-agent CEM and the monotonic nonlinear critic decomposition. This ablation study aims to disclose the impact of replacing the non-linear decomposition with the linear decomposition as in (Wang et al., 2020b). As shown in fig. 4, the performance of the linear decomposition, denoted as *MCEM_NCD_linear*, is comparable to that of the nonlinear decomposition in the easier scenarios. But the nonlinear decomposition outperforms significantly in the complicated scenarios, including two *Super Hard* scenarios in discrete action tasks and the 9_agents_and_3_preys scenario in continuous action tasks.

*Off-policy vs. On-policy*: MCEM-NCD employs an $n$-step return-based off-policy approach with Sarsa and Retrace to estimate the update target of the centralized critic. To assess the effectiveness of this design, we construct an on-policy variant, denoted *MCEM-NCD_on-policy*, which replaces the off-policy target with the $n$-step return-based on-policy Sarsa update (Retrace is excluded, as it is specific to off-policy learning). As

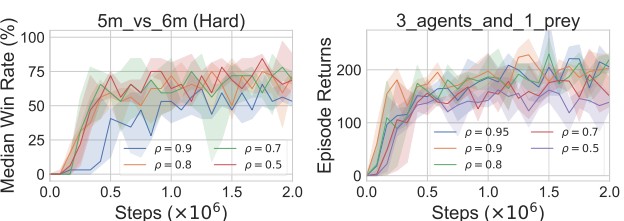

Figure 5: Impact of $\rho$.

shown in fig. 4, MCEM-NCD consistently dominates *MCEM-NCD_on-policy* across all scenarios for both discrete and continuous action tasks.

*Retrace vs TB*: When learning stochastic policies with discrete actions in (Wang et al., 2020b), the $n$-step return-based off-policy with Expected Sarsa and TB where linear critic decomposition is applied. Due to the nonlinear critic decomposition, MCEM-NCD cannot be adapted by replacing Sarsa with Expected Sarsa. But we can replace Retrace with TB in MCEM-NCD, denoted as *MCEM-NCD_tree-backup*, to learn whether Retrace is better than TB. As shown in fig. 4, MCEM-NCD consistently outperforms MCEM-NCD_tree-backup across all scenarios, with the advantage being particularly pronounced in complex environments.".

### 6.4 IMPACT OF $\rho$

The multi-agent CEM updates agent policies using joint action samples from the top $(1 - \rho)$ quantile ranked by the joint action-value $Q_{tot}$. We evaluate how different choices of $\rho$ affect performance. As shown in fig. 5, performance is sensitive to this parameter. A larger $\rho$ reduces the number of samples available for policy updates, lowering estimation accuracy and potentially impairing learning stability. In contrast, a smaller provides more samples but increases the risk of reinforcing suboptimal actions, i.e., assigning higher probabilities to actions that should remain unlikely.

## 7 CONCLUSION

This study advances cooperative multi-agent reinforcement learning by introducing MCEM-NCD, a novel method that reconciles the strengths of nonlinear decomposition with the need to mitigate the centralized–decentralized mismatch. By extending the Cross-Entropy Method to the multi-agent setting, MCEM-NCD enables policy updates that exclude suboptimal joint actions, ensuring more robust and stable learning across agents. Furthermore, our integration of Sarsa with Retrace optimizes off-policy learning under nonlinear decomposition, improving sample efficiency without sacrificing computational tractability. Empirical evaluations across benchmark environments confirm that MCEM-NCD consistently outperforms state-of-the-art approaches in both discrete and continuous action spaces. These findings highlight the promise of MCEM-NCD as a scalable and expressive framework for cooperative MARL, paving the way for future research on efficient and reliable coordination among multiple agents in complex environments.

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

## A    PROOF OF THEOREM 5.1

*Proof.* Given a set of joint actions $E(\boldsymbol{\tau})$ for $\boldsymbol{\tau}$, MCEM-NCD increases the probability of joint actions in $I(\boldsymbol{\tau}) \subset E(\boldsymbol{\tau})$, which are the top $(1-\rho)$ quantile according to $Q_{tot}^{\boldsymbol{\pi}_g}(\boldsymbol{\tau}, \cdot)$. That is, each agent $a$ increases $\pi_\rho^a(u^a|\tau^a)$ of action $u^a \in \boldsymbol{u}$ for every joint action $\boldsymbol{u} \in I(\boldsymbol{\tau})$. In contrast, the method with centralized gradient increases the probability of joint action $\boldsymbol{u}$ if $Q_{tot}^{\boldsymbol{\pi}_g}(\boldsymbol{\tau}, \boldsymbol{u})$ is high, and decreases it if $Q_{tot}^{\boldsymbol{\pi}_g}(\boldsymbol{\tau}, \boldsymbol{u})$ is low. The low $Q_{tot}^{\boldsymbol{\pi}_g}(\boldsymbol{\tau}, \boldsymbol{u})$ may be caused by the suboptimal behavior of some agents rather than all agents. So, although the behavior of an agent $a$ is optimal, it still needs to decrease $\pi_g^a(u^a|\tau^a)$. If the optimal behavior of the agent $a$ is a part of another joint action $\boldsymbol{u}'$ where $Q_{tot}^{\boldsymbol{\pi}_g}(\boldsymbol{\tau}, \boldsymbol{u}')$ is high, it reduces the probability of joint action $\boldsymbol{u}'$. So, the following relations hold:

$$\sum_{\boldsymbol{u} \in \boldsymbol{U}} \boldsymbol{\pi}_g(\boldsymbol{u}|\boldsymbol{\tau})Q_{tot}^{\boldsymbol{\pi}_g}(\boldsymbol{\tau}, \boldsymbol{u}) = \mathbb{E}_{\boldsymbol{\pi}_g}[Q_{tot}^{\boldsymbol{\pi}_g}(\boldsymbol{\tau}, \boldsymbol{u})]$$

$$\leq \mathbb{E}_{\boldsymbol{\pi}_\rho}[Q_{tot}^{\boldsymbol{\pi}_g}(\boldsymbol{\tau}, \boldsymbol{u})]$$
$$= \mathbb{E}_{\boldsymbol{\pi}_\rho}\{r_{t+1} + \gamma\mathbb{E}_{\boldsymbol{\pi}_g}[Q_{tot}^{\boldsymbol{\pi}_g}(\boldsymbol{\tau}_{t+1}, \boldsymbol{u}_{t+1})]|_{\tau_t=\boldsymbol{\tau}, u_t=\boldsymbol{u}}\}$$
$$\leq \mathbb{E}_{\boldsymbol{\pi}_\rho}\{r_{t+1} + \gamma\mathbb{E}_{\boldsymbol{\pi}_\rho}[Q_{tot}^{\boldsymbol{\pi}_g}(\boldsymbol{\tau}_{t+1}, \boldsymbol{u}_{t+1})]|_{\tau_t=\boldsymbol{\tau}, u_t=\boldsymbol{u}}\}$$
$$= \mathbb{E}_{\boldsymbol{\pi}_\rho}\{r_{t+1} + \gamma r_{t+2} + \gamma^2\mathbb{E}_{\boldsymbol{\pi}_g}[Q_{tot}^{\boldsymbol{\pi}_g}(\boldsymbol{\tau}_{t+2}, \boldsymbol{u}_{t+2})]|_{\tau_t=\boldsymbol{\tau}, u_t=\boldsymbol{u}}\}$$
$$\leq \mathbb{E}_{\boldsymbol{\pi}_\rho}\{r_{t+1} + \gamma r_{t+2} + \gamma^2\mathbb{E}_{\boldsymbol{\pi}_\rho}[Q_{tot}^{\boldsymbol{\pi}_g}(\boldsymbol{\tau}_{t+2}, \boldsymbol{u}_{t+2})]|_{\tau_t=\boldsymbol{\tau}, u_t=\boldsymbol{u}}\}$$
$$\dots$$
$$\leq \mathbb{E}_{\boldsymbol{\pi}_\rho}\{r_{t+1} + \gamma r_{t+2} + \gamma^3 r_{t+3} + \dots + \gamma^3 r_T|_{\tau_t=\boldsymbol{\tau}, u_t=\boldsymbol{u}}\}$$
$$= \mathbb{E}_{\boldsymbol{\pi}_\rho}[Q_{tot}^{\boldsymbol{\pi}_g}(\boldsymbol{\tau}, \boldsymbol{u})] = \sum_{\boldsymbol{u} \in \boldsymbol{U}} \boldsymbol{\pi}_\rho(\boldsymbol{u}|\boldsymbol{\tau})Q_{tot}^{\boldsymbol{\pi}_\rho}(\boldsymbol{\tau}, \boldsymbol{u})$$

The inequality in eq. (14) is proved. In a similar way, the inequality for continuous actions in eq. (15) can be proved.

$\square$

## B    STOCHASTIC POLICIES WITH CONTINUOUS ACTIONS-GAUSSIAN DISTRIBUTION

The policy $\pi(u|\tau; \theta)$ is this parameterized probability distribution (Sutton & Barto, 2018):

$$\pi(\tau; \theta) \equiv \mathcal{N}(\mu(\tau; \theta), \sigma(\tau; \theta)^2)$$

Given $\tau$, the network outputs the distribution parameters $\mu$ and $\sigma$. The policy is then:

$$\pi(u|\tau; \theta) = \mathcal{N}(u|\mu, \sigma^2) = \frac{1}{\sigma\sqrt{2\pi}} \exp\left(-\frac{(u-\mu)^2}{2\sigma^2}\right)$$

We replace $\pi(u|\tau; \theta)$ with the probability density function of the Gaussian distribution:

$$\log \pi(u|\tau; \theta) = \log\left[\mathcal{N}(u|\mu, \sigma^2)\right]$$

Substitute into the formula for the Gaussian distribution:

$$\log \pi(u|\tau; \theta) = \log\left(\frac{1}{\sigma\sqrt{2\pi}} \exp\left(-\frac{(u-\mu)^2}{2\sigma^2}\right)\right)$$

We replace $\pi(\cdot|\tau;\theta)$ with the Gaussian distribution and substitute into the formula for the entropy of the Gaussian distribution (Shannon, 1948):

$$
\begin{aligned}
\mathcal{H}(\pi(\cdot|\tau;\theta)) &= \mathcal{H}(\mathcal{N}(\cdot|\mu,\sigma^2)) \\
&= -\mathrm{E}_{u\in U}\left[\log\left[\mathcal{N}(u|\mu_i,\sigma_i^2)\right]\right] \\
&= -\mathrm{E}_{u\in U}\left[\log\left(\frac{1}{\sigma_i\sqrt{2\pi}}\exp\left(-\frac{(u-\mu)^2}{2\sigma^2}\right)\right)\right] \\
&= -\mathrm{E}_{u\in U}\left[-\frac{1}{2}\ln(2\pi\sigma_i^2)-\frac{1}{2}\left(\frac{u-\mu}{\sigma_i}\right)^2\right] \\
&= \frac{1}{2}\log(2\pi\sigma_i^2)+\frac{1}{2}\mathrm{E}_{u\in U}\left[\left(\frac{u-\mu}{\sigma_i}\right)^2\right] \\
&= \frac{1}{2}\log(2\pi\sigma_i^2)+\frac{\mathrm{E}_{u\in U}\left[(u-\mu)^2\right]}{2\sigma^2} \\
&= \frac{1}{2}\log(2\pi\sigma^2)+\frac{\sigma^2}{2\sigma^2} \\
&= \frac{1}{2}\log(2\pi\sigma_i^2)+\frac{1}{2}\log(e) \\
&= \frac{1}{2}\log(2\pi\sigma_i^2 e)
\end{aligned}
$$

## C PSEUDOCODE OF MCEM-NCD

---
**Algorithm 1** MCEM-NCD

---
1: Randomly initialize parameters $\psi$, $\phi = \{\phi^a\}_{a\in A}$, $\boldsymbol{\theta} = \{\theta^a\}_{a\in A}$, and $\hat{\boldsymbol{\theta}} = \{\hat{\theta}^a\}_{a\in A}$ for the critic decomposition network NCD, action-value functions $\boldsymbol{Q} = \{Q^a\}_{a\in A}$, the main policies $\boldsymbol{\pi} = \{\pi^a\}_{a\in A}$, and the proposal policies $\hat{\boldsymbol{\pi}} = \{\hat{\pi}^a\}_{a\in A}$, respectively; initialize a replay buffer $\mathcal{D} = \emptyset$;

2: **for** 1 to $T$ **do**

3:     Trajectories generated where each agent $a$ follows its current policy $\pi^a_{a\in A}$ and stored in $\mathcal{D}$;

4:     Sample a batch $B$ of trajectories from $\mathcal{D}$;

5:     Update parameters related to centralized critic, $\psi$ and $\phi$, by minimizing loss: (eq. (12)):

6:     $\mathcal{L}(\psi,\phi) = \mathbb{E}_{(\boldsymbol{\tau},\boldsymbol{a})\in B}\left[(\mathcal{R}Q_{tot}(\boldsymbol{\tau},\boldsymbol{a})-Q_{tot}(\boldsymbol{\tau},\boldsymbol{a};\psi,\phi))^2\right]$

7:     **for** each record in the trajectories of $B$, extract $\boldsymbol{\tau}$ **do**

8:         Sample a set of joint actions $E(\boldsymbol{\tau})$ where each $\boldsymbol{u} = \{u^a\}_{a\in A}$ and $u^a \sim \hat{\pi}^a(\cdot|\tau^a)$;

9:         Compute $Q_{tot}(\boldsymbol{\tau},\boldsymbol{u})$ for each joint action $\boldsymbol{u} \in E(\boldsymbol{\tau})$;

10:        Identify $I(\boldsymbol{\tau})$ including all $\boldsymbol{u} \in E(\boldsymbol{\tau})$ in the top $(1-\rho)$ quantile according to $Q_{tot}(\boldsymbol{\tau},\cdot)$;

11:     **end for**

12:     Updating main and proposal policies by ascending gradients (eq. (6) and eq. (7)):

13:     $\nabla J(\boldsymbol{\theta}) = \sum_{\boldsymbol{\tau}\in B}\sum_{\boldsymbol{u}\in I(\boldsymbol{\tau})}\sum_{a\in A}\nabla_{\theta^a}\ln\pi^a(u^a|\tau^a;\theta^a)$

14:     $\nabla J(\hat{\boldsymbol{\theta}}) = \sum_{\boldsymbol{\tau}\in B}\sum_{\boldsymbol{u}\in I(\boldsymbol{\tau})}\sum_{a\in A}\nabla_{\hat{\theta}^a}\ln\hat{\pi}^a(u^a|\tau^a;\hat{\theta}^a)+\beta\nabla_{\hat{\theta}^a}\mathcal{H}\left(\hat{\pi}^a(\cdot|\tau^a;\hat{\theta}^a)\right)$

15: **end for**

---

## D ROLE OF LLMS

LLMs have been used to polish writing only in this paper. We would like to clarify that LLMs do not play any role in research ideation and/or writing to the extent that they could be regarded as a contributor.

