# OpenReview forum: "Multi-Agent Cross-Entropy Method with Monotonic Nonlinear Critic Decomposition"
_ICLR.cc/2026/Conference — ICLR 2026 Conference Withdrawn Submission_

### Official Review · Reviewer_JGhA · 2025-10-19

**Soundness:** 2
**Presentation:** 3
**Contribution:** 2
**Rating:** 2
**Confidence:** 4

**Summary:**

This paper extends the Cross-Entropy Method (CEM) to multi-agent settings and combines it with a monotonic nonlinear critic decomposition (NCD) to enhance representational power while mitigating the centralized–decentralized mismatch (CDM).
To improve sample efficiency, the authors employ Sarsa + Retrace to construct off-policy targets for training the decomposed critic.
Experimental results suggest that the proposed method can outperform existing policy-based baselines.

**Strengths:**

- Extending CEM to multi-agent settings and pairing it with a monotonic nonlinear decomposition is an interesting and novel direction.

- The approach demonstrates promising preliminary results and could inspire future work on cross-entropy–based policy optimization in cooperative MARL.

**Weaknesses:**

1. The theoretical analysis is informal and lacks rigor.
In particular, I do not see why the first inequality in Appendix A, $E_{\pi_g}[Q^{\pi_g}]\le E_{\pi_\rho}[Q^{\pi_g}]$,
should hold under the description provided.
The authors should formally define both $\pi_g$ and $\pi_\rho$.
Are they final converged policies, or intermediate policies after one policy improvement step?
The proof also implicitly assumes access to the exact $Q^\pi$, whereas in practice $Q^\pi$ is estimated by a QMIX-style network with monotonic constraints that induce bias.
Therefore, it is unclear whether the proposed method guarantees true policy improvement.

2. In section 4.2, the paper states that $E_\pi[Q_{tot}(\tau_{t+1},\cdot)]$ is replaced by $Q_{tot}(\tau_{t+1},u_{t+1})$ with $u_{t+1}\sim\pi$.
However, by definition these quantities are equivalent—the latter is simply a Monte Carlo sample of the former—and this substitution does not introduce any new idea.
Furthermore, simply adopting Retrace from prior work is not novel.
More importantly, there is no theoretical or empirical discussion of whether Retrace’s convergence or bias-control properties still hold under a nonlinear decomposition and function approximation, as used in NCD.

3. MCEM samples joint actions by independently sampling from each agent’s policy to form a set $E(\tau)$, which is then ranked by $Q_{tot}$ to select elites.
However, as the number of agents grows, the joint-action space increases exponentially.
Using a fixed small number of joint samples (10 for discrete, 20 for continuous, as stated in the experiments) is unlikely to capture meaningful coordinated behaviors.
In high-dimensional or continuous-action settings, a small $E(\tau)$ will likely miss high-value joint actions, and CEM may collapse to suboptimal modes.

4. Several configuration details raise significant concerns about fairness and reproducibility:

- The paper does not list hyperparameters in an appendix, forcing readers to locate them manually.

- In the MCEM-NCD configuration, the parameter batch_size_run is set to 1, whereas the default in PyMARL2 is 8. This discrepancy can lead to substantial performance differences.

- The critic_hidden_dim is set to 256, which is larger than that used by many baselines such as RIIT (128), possibly inflating performance.

  Such inconsistencies should be clearly justified, and all experimental hyperparameters should be fully reported.

5. It remains unclear why the Cross-Entropy Method is an appropriate or necessary tool for addressing the centralized–decentralized mismatch (CDM).
The proposed method appears to make QMIX “on-policy” by introducing a CEM-based actor search, and then convert it back to off-policy training via Retrace corrections.
This two-step design seems ad hoc and raises the question: why not directly use an off-policy actor-critic formulation guided by the centralized critic?
The paper should better justify how CEM fundamentally helps resolve CDM, rather than serving as a heuristic joint-action search.

**Questions:**

1. The policy update in Eq. (6)–(7) maximizes the likelihood over elites.
Is this update biased compared to a standard policy gradient?
Please clarify the relationship between your update and REINFORCE or advantage-weighted regression.

2. What is the computational overhead of MCEM relative to a standard centralized policy-gradient approach?

---

### Official Review · Reviewer_P8CF · 2025-10-29

**Soundness:** 3
**Presentation:** 2
**Contribution:** 2
**Rating:** 2
**Confidence:** 4

**Summary:**

This work introduces a MA cross-entropy method, combined with monotonic nonlinear critic decomposition, in order to address the issue of centralised-decentralised mismatch caused by suboptimal behaviours.

**Strengths:**

1. The extension of the CEM form the single-agent to MARL setting is natural and well-formulated as a percentile-greedy policy.
2. The computational details of the experiments are sufficient.
3. The results in SMAC against related VD methods are somewhat convincing, as MCEM NCD performs better or the same as other baselines (see below).

**Weaknesses:**

1. The primary implementation contribution seems minimal - the only difference appears to be in how the actions are selected for the fit of the network. While Theorem 5.1 appears to demonstrate that the MCEM method should perform at least as well as baseline, this doesn't guarantee improvement in the general setting. Why was no equilibrium analysis or spectral analysis of the game dynamics with factorization performed to motivate the method further? Indeed, in the results, MCEM sometimes performs within the statistical margins of other baselines.
2. What is the shaded area in Figure 2? We need to know this in order to assess the statistical significance of the method.
3. While SMAC and PP are relevant benchmarks, additional benchmarks would supplement this work further as they are not considered competitive MARL benchmarks anymore (SMAC can even be solved by open-loop policies[1]). The authors could consider Mamujoco, SMACv2, Overcooked, or any of the common reward benchmarks in Benchmarl [2].
4. While deep MARL benchmarks are useful, a didactic example of centralised-decentralised mismatch would be pivotal in motivating this work. Is there a game which requires the cross-entropy sampling to be solvable? Even a contrived setting would motivate the usefulness of the proposed method.

[1] Ellis, Benjamin, et al. "Smacv2: An improved benchmark for cooperative multi-agent reinforcement learning." Advances in Neural Information Processing Systems 36 (2023): 37567-37593.
[2] Bettini, Matteo, Amanda Prorok, and Vincent Moens. "Benchmarl: Benchmarking multi-agent reinforcement learning." Journal of Machine Learning Research 25.217 (2024): 1-10.

**Questions:**

1. Can the MCEM method be generalized to non-monotonic decomposition methods such as FACMAC?

---

### Official Review · Reviewer_UGSh · 2025-10-31

**Soundness:** 1
**Presentation:** 2
**Contribution:** 1
**Rating:** 2
**Confidence:** 4

**Summary:**

This paper proposes a new multi-agent reinforcement learning algorithm based on the cross-entropy method to solve the centralized–decentralized mismatch issue in MARL. Simulation results demonstrate the effectiveness of the proposed method.

**Strengths:**

Simulations are conducted on standard MARL benchmarks, and advanced baselines are compared.

**Weaknesses:**

1. Many policy gradient formulas in this paper are incorrect.

2. The use of the auxiliary proposal policies is unclear.

3. The convergence of the proposed algorithm cannot be guaranteed from a theoretical perspective.

4. For the continuous action setting, the authors are recommended to evaluate their proposed algorithm on benchmarks with high-dimensional action spaces, and to discuss the hyperparameter settings related to the cross-entropy method in their algorithm.

**Questions:**

None.

---

### Official Review · Reviewer_WPdG · 2025-11-01

**Soundness:** 3
**Presentation:** 2
**Contribution:** 2
**Rating:** 4
**Confidence:** 4

**Summary:**

This paper aims to develop a policy gradient method using nonlinear value function decomposition in collaborative MARL, avoiding the CDM problem under the CTDE framework. The authors extend CEM from single-agent to multi-agent scenarios, using Q-total to quantile-select sampled joint actions, intending to use these elite actions to update the decentralized policy network of each agent. This mechanism alleviates the CDM problem by rejecting suboptimal joint actions and eliminating the suboptimal influence of some agents. To improve sample efficiency, the authors designed an off-policy Critic learning method, modifying the k-step reward objective based on Expected Sarsa in DOP to a Sarsa-based form, thus ensuring compatibility with nonlinear decomposition. Simultaneously, the Retrace algorithm is introduced to replace the traditional TB and IS to reduce variance and improve learning stability.

In the experimental section, the authors validated the effectiveness of the algorithm on the SMAC benchmark in the discrete actions space and in the Predator-Prey environment in the continuous actions space. The results show that MCEM-NCD significantly outperforms existing state-of-the-art methods in both convergence speed and final performance.

**Strengths:**

This paper uses CEM to "filter" suboptimal actions to avoid centralized gradients, serving as an alternative to traditional centralized critic policy gradient methods and offering insights for large-scale agent tasks.

The combination of MCEM and NCD is seamless, preserving both the excellent expressive power of nonlinear decomposition and the high quality of updated data. Furthermore, the improvements to the heterogeneous policy critic learning part (Sarsa + Retrace) are well-considered, enhancing the algorithm's practicality and stability.

Effective experimental validation was conducted on discrete and continuous control tasks, particularly demonstrating significant improvements in difficult and ultra-difficult SMAC scenarios. Comparisons with a series of representative state-of-the-art methods, such as DOP, FACMAC, VDAC-mix, and LICA, lend high credibility to the experimental results. Ablation studies clearly demonstrate the contribution of each component, strongly supporting the authors' design choices.

**Weaknesses:**

W1: Theorem 5.1 and its proof in Appendix A form the core theoretical analysis of this paper. However, the proof is overly simplistic and intuitive, lacking rigorous mathematical form. It reads more like a description of the algorithm's design intent—to improve expected returns by selecting actions with high Q values—than a rigorous mathematical proof. Why does the first inequality $E_{\pi_g}[Q_{\pi_g}^{tot}(\tau, u)] \le E_{\pi_\rho}[Q_{\pi_g}^{tot}(\tau, u)] $ hold? Subsequent recursive expansions all rely on this condition. This significantly undermines the reliability of the conclusion that percentile-greedy strategies are at least as good as centered gradient strategies.

W2: The FACMAC paper states that a centralized critic gradient is used to avoid the incoordination problem of individual gradient updates. This is a design choice and does not mean that individual $Q_i$ cannot be used as a critic to guide gradient updates. If the authors claim this is a fundamental drawback, they should design appropriate toy examples to illustrate this property, such as whether MCEM can solve the failure case in Figure 3 of FACMAC.

W3: Using MCEM can partially alleviate the requirement for accuracy of the centralized critic through sampling, which is necessary for tasks with a larger number of agents, where it is often difficult to learn an effective centralized critic. However, as the number of agents k increases, the joint action space grows exponentially. Is it still possible to find effective "elite" joint actions with a small amount of sampling? If all sampled data is trapped in local optima or poor data, can the algorithm still guarantee certain performance? The paper lacks discussion on this scalability issue.

W4: This article has many inaccuracies. It would be better to replace "records" with "transitions". In line 296, a parenthesis is missing. In line 300, it should be Fig. 1. In line 472, it should be "smaller $\rho$".

W5: GitHub links should be anonymous to comply with review guidelines.

**Questions:**

Q1: The exploration is controlled by the proposal policies with entropy regularization. However, how to guarantee the consistency between proposal policies and main policies? If there exists multiple optimal modes in the proposal policies' samples, will it lead to mode collapse or to oscillate between multiple behavioral patterns, thus affecting the stability of learning?

Q2: Can we directly use Q total to filter buffer data to obtain a higher quality centralized critic? Before Q total converges, how can we ensure that the data selected is accurately ranked by relative value? How does the algorithm prevent early-stage Q-function errors from causing catastrophic policy collapse?

Q3: The predator-prey task has a very low dimension of state and action. Is it feasible to control the task in a higher dimension, such as MAMujoco?

Other problems mentioned in the weaknesses.

---

### Note · Authors · 2025-11-23

I have read and agree with the venue's withdrawal policy on behalf of myself and my co-authors.